# Interval walking training in type 2 diabetes: A pilot study to evaluate the applicability as exercise therapy

**Kouhei Kitajima[1], Ako Oiwa[1]\*, Takahiro Miyakoshi[1], Manami Hosokawa[1], Mayuka Furihata[2], Masaaki Takahashi[3], Shizue Masuki[2], Hiroshi Nose[2], Yosuke Okubo[1], Ai Sato[1], Masanori Yamazaki[1], Mitsuhisa Komatsu[1]**

**1** Division of Internal Medicine, Department of Diabetes, Endocrinology and Metabolism, Shinshu University School of Medicine, Matsumoto, Nagano, Japan, **2** Department of Sports Medical Sciences, Shinshu University Graduate School of Medicine, Matsumoto, Nagano, Japan, **3** Department of Radiology, Shinshu University Graduate School of Medicine, Matsumoto, Nagano, Japan

\* akooiwa@shinshu-u.ac.jp

**Data Availability Statement:** All relevant data are within the paper and its Supporting Information files.

## Abstract

There are few established easy-to-perform exercise protocols with evidence-based effects for individuals with type 2 diabetes (T2D). A unique exercise regimen, interval walking training (IWT), has been reported to be beneficial for improving metabolic function, physical fitness and muscle strength in adults of overall health. This pilot study aims to demonstrate descriptive statistics of IWT adherence and changes in various data before and after the intervention of IWT in adults with T2D, perform statistical hypothesis testing, and calculate effect sizes. We performed a single-arm interventional pilot study with IWT for 20 weeks. We enrolled 51 participants with T2D aged 20–80 years with glycohemoglobin (HbA1c) levels of 6.5–10.0% (48–86 mmol/mol) and a body mass index of 20–34 kg/m$^2$, respectively. The target was 60 min/week of fast walking for 20 weeks. The participants visited the hospital and were examined at 4-week intervals during this period. Between the start of IWT and after 20 weeks, we measured and evaluated changes in glucose and lipid metabolism data, body composition, physical fitness, muscle strength, dietary calorie intake, and daily exercise calories. All included participants completed IWT, with 39% of them reaching the target length of fast walking over 1,200 minutes in 20 weeks. In the primary outcome, HbA1c levels, and in the secondary, lipid metabolism and body composition, no significant changes were observed except for high-density lipoprotein cholesterol (HDL-C) (from 1.4 mmol/L to 1.5 mmol/L, $p = 0.0093$, t-test). However, in the target achievement group, a significant increase in VO$_2$ peak by 10% (from 1,682 mL/min to 1,827 mL/min, $p = 0.037$, t-test) was observed. Effect sizes were *Cohen's d* = 0.25 of HDL-C, -0.55 of triglyceride, and 0.24 of VO$_2$ peak in the target achievement group, which were considered to be of small to medium clinical significance. These results could be solely attributed to IWT since there were no significant differences in dietary intake and daily life energy consumption before and after the study. IWT could be highly versatile and was suggested to have a positive effect on lipid metabolism and physical fitness. In future randomized controlled trial (RCT) studies, the detailed effects of IWT, focusing on these parameters, will be examined.

**Funding:** The authors received no specific funding for this work.

**Competing interests:** The authors have declared that no competing interests exist.

**Trial registration:** This trial was registered with the Japanese University Hospital Medical Information Network Clinical Trials Registry (UMIN-CTR: Usefulness on interval walking training in patients with type 2 diabetes. 000037303).

## Introduction

The International Diabetes Federation estimated 450 million people with diabetes worldwide in 2019 [1]. Exercise therapy and diet therapy form the basis of diabetes treatment and are required for most people with type 2 diabetes (T2D). In recent years, combined aerobic and resistance exercises have reportedly enhanced muscle mass and strength, resulting in improved insulin resistance and more effective glycemic control [2–4] compared to aerobic exercise alone [5]. Accordingly, there are worldwide recommendations for performing both aerobic and resistance exercises [6].

On the other hand, there is a low uptake of exercise, with 60% of adults with T2D not exercising at all [7]. This low practice rate could be attributed to low motivation, limited time availability, management problems, and lack of willpower or control [8]. Another reason for the low exercise uptake among people with T2D could be difficulty on the part of medical professional to teach positive habits, given the few established easy-to-perform protocols with evidence-based effects [9].

The Department of Sports Medical Sciences, Shinshu University Graduate School of Medicine, developed a unique walking method termed interval walking training (IWT) in 1999. Interval walking involves repeating fast walking at $\geq 70\%$ of the individual peak aerobic capacity ($VO_2$ peak) and slow walking at $\leq 40\%$ $VO_2$ peak alternately for 3 minutes. Target exercise amounts are $\geq 60$ minutes per week of fast walking time, which means 5–10 sets of 3 minutes of fast walking and 3 minutes of the slow walking a day for $\geq 4$ days. This was determined by the effect of increasing $VO_2$ peak almost plateaued at 50 minutes per week [10]. Further, the aforementioned department also developed a triaxial accelerometer called JD Mate (Kissei Comtec, Matsumoto, Japan) that can easily estimate $VO_2$ peak [11]. Details of the IWT are published elsewhere [12, 13]. There has been active research and development of IWT, with approximately 8,700 individuals of overall good health participating in these studies [13, 14]. Despite walking being a simple exercise, IWT improves metabolic function, physical fitness and muscle strength, indicating that it combines aerobic and resistance exercises. There is a plethora of scientific evidence supporting the efficacy of IWT, with increasing worldwide attention [10, 12, 13, 15–17].

Currently, only one study has investigated IWT using JD Mate in persons with T2D; compared with continuous walking, IWT improved physical fitness, body composition, and glycemic control [18]. However, because the basis for the number of cases required for statistical analysis in this study is unclear, it is difficult to accurately judge the effect of IWT on patients with T2D from this study alone. In addition, the participants were limited to a small number who could strictly follow the protocol, which rendered it insufficient to determine the general and pragmatic efficacy of IWT for T2D.

This single-arm intervention study plays an exploratory role in accurately assessing the practical effectiveness of IWT with T2D.

## Materials and methods

The protocol for this research project has been approved by the institution's suitably constituted Ethics Committee. It conforms to the provisions of the Declaration of Helsinki.

The clinical study protocol was approved by the ethics committee of the Shinshu University School of Medicine (No.4374). Written informed consent was obtained from all participants before participation. This trial was registered in the Japanese University Hospital Medical Information Network Clinical Trials Registry (UMIN-CTR: Usefulness on interval walking training in patients with T2D. UMIN 000037303).

## Research design

This single-arm interventional study was conducted from July 1, 2019 to December 31, 2020. Results from the study are reported in line with the Consolidated Standards of Reporting Trials: CONSORT. The flow of participants is presented in Fig 1.

Fig 2 shows a flow chart that simplifies this study method. Participants performed IWT for 20 weeks. The target fast walking time was set to $\geq$ 60 minutes per week, which means 5–10 sets of 3 minutes of fast walking and 3 minutes of slow walking a day for $\geq$ 4 days. The target fast walking time was set to $\geq$ 60 minutes per week due to the report that the effect of increasing VO$_2$ peak almost plateaued at 50 minutes per week [10]. Moreover, the reason for the setting of 20 weeks is that we have found that physical fitness improvement with IWT occurs after 20 weeks of intervention (12), and physical fitness increases by 10–15% in the first 20 weeks and then maintains it with continued IWT (13). We have set the primary outcome as a change in HbA1c levels, and the secondary outcome as a change in body composition, physical fitness, muscle strength, dietary calorie intake, and daily exercise calories, before and after IWT.

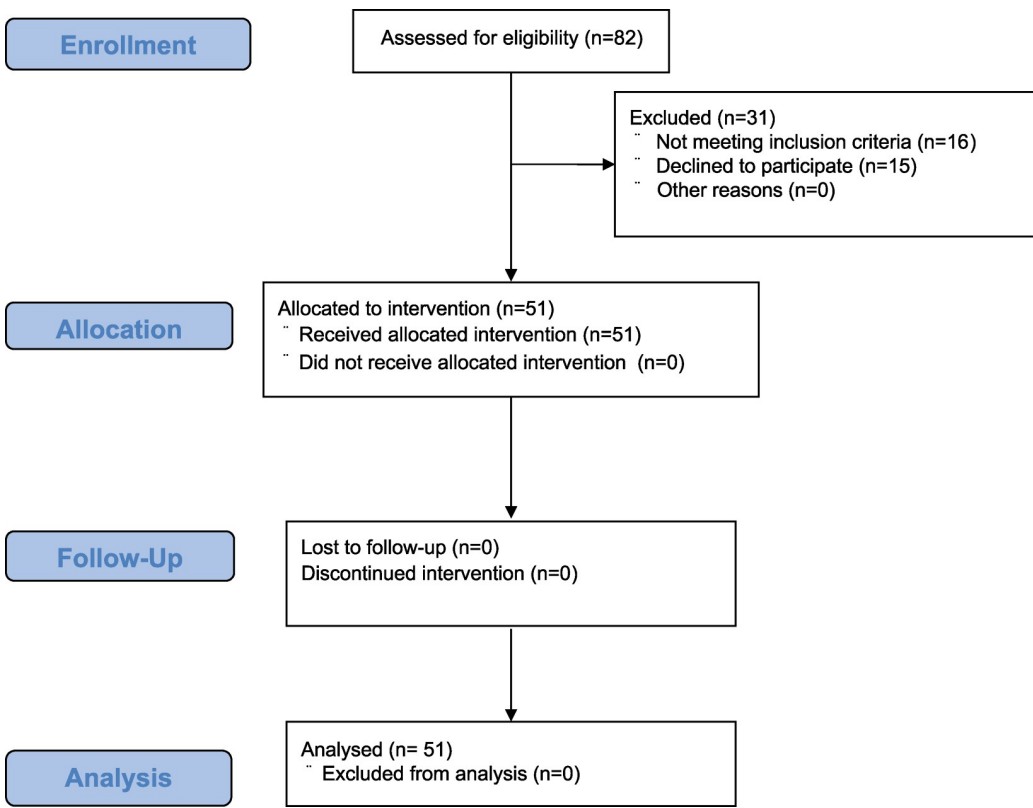

**Fig 1. Flow chart of the progress through the phases of the study.** This is a single-arm interventional study. All participants received the intervention, and their data were analyzed.

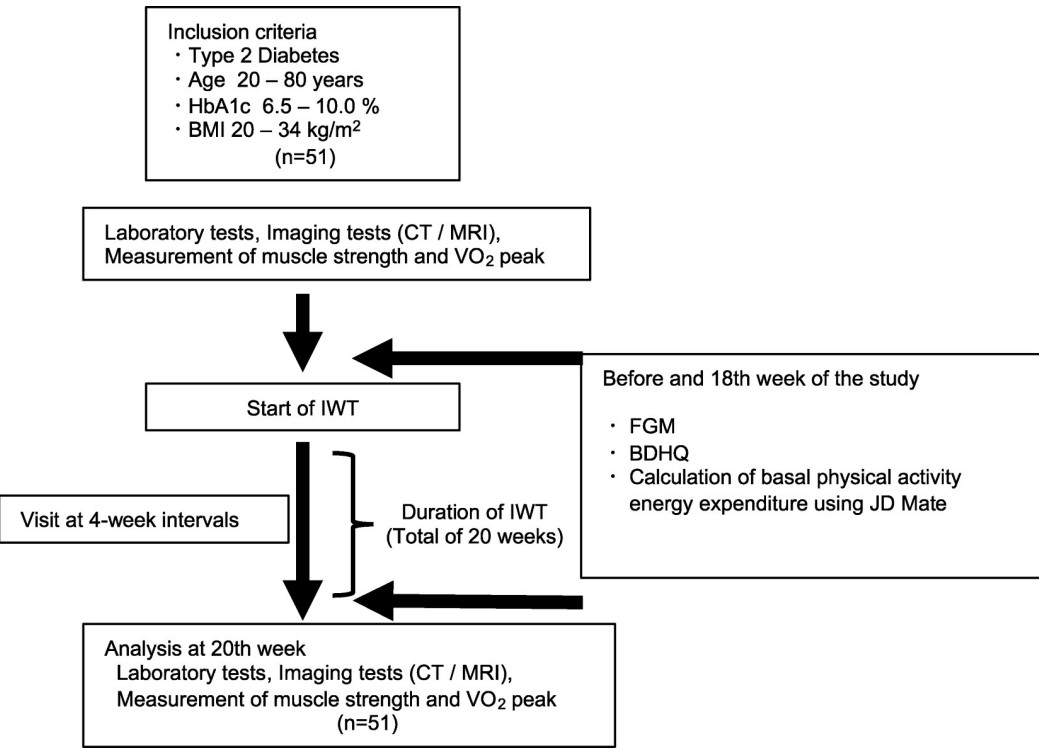

**Fig 2. Flow chart of this research method.** Fifty-one participants met all inclusion criteria and none were excluded. Both before starting and after IWT, participants underwent laboratory tests and imaging tests (computed tomography (CT) / magnetic resonance imaging (MRI)) and were measured for muscle strength and VO₂ peak. Flash glucose monitoring (FreeStyle Libre pro, ABBOTT Japan Inc. Chiba, Japan, FGM), brief self-administered diet history questionnaire (BDHQ), and calculation of basal physical activity energy expenditure examination were performed both for 1 to 2 weeks before the study and for 1 to 2 weeks from the 18th week of the study. The total study period was 20 weeks, and the participants visited the hospital and were examined at 4-week intervals during the period. BMI, body mass index; CT, computed tomography; MRI, magnetic resonance imaging; IWT, interval walking training; FGM, flash glucose monitoring; BDHQ, brief self-administered diet history questionnaire.

## Participants

Recruitment of participants included all outpatients who visited the Department of Diabetes, Endocrinology and Metabolism, Shinshu University Hospital, from July 1, 2019, to December 31, 2019, and met all the inclusion criteria; none of them were excluded. The inclusion criteria were having T2D; age: 20–80 years; HbA1c: 6.5–10.0% (48–86 mmol/mol); body mass index (BMI): 20–34 kg/m², and capability to perform exercise therapy as per the exercise regimen. In addition, we excluded persons with pre-proliferative or proliferative diabetic retinopathy, diabetic nephropathy with albuminuria ($\geq$300 mg/gCre), or an estimated glomerular filtration rate (eGFR) < 30 mL/min/1.73 m² calculated by serum creatinine, history of coronary artery disease or stroke, pregnant or lactating women, and cases judged by the doctors as inappropriate.

## Outcome measures

Changes in the following parameters were measured and evaluated between the start of IWT and after 20 weeks.

Laboratory tests: HbA1c, plasma glucose levels, plasma C peptide, immunoreactive insulin (IRI), low-density lipoprotein (LDL) cholesterol, high-density lipoprotein (HDL) cholesterol,

triglyceride, blood urea nitrogen (BUN), serum creatinine, aspartate aminotransferase, alanine aminotransferase, urinary albumin excretion.

Imaging tests: MRI-quantified plasma liver fat mass, CT-quantified abdominal visceral fat mass and thigh muscle mass.

Physical fitness tests: Thigh muscle strength measured using a dynamometer (Biodex 3, Biodex Medical System, Shirley, NY, USA), $VO_2$ peak (maximum) using JD Mate.

Other parameters: BMI, systolic/diastolic blood pressure, prescription, frequency of interval walking, target achievement rate, mean amplitude of glycemic excursions (MAGE) measured using FGM, dietary calorie intake using BDHQ and recorded daily exercise calories burned using JD Mate.

## Intervention protocol

A schematic of the intervention protocol is shown in Fig 2.

Before starting IWT, background characteristics (family history, drinking, smoking, and exercise habits) were collected using a questionnaire; the participants underwent abdominal and thigh plain CT and abdominal MRI. Moreover, we measured the $VO_2$ peak and lower limb muscle strength following the physical fitness measurement protocol at the Jukunen Taii-kudaigaku Research Center (JTRC) [10]. Further, a FGM device was worn for two weeks before starting IWT. The dietary survey undergone with BDHQ [19], and calories burned in daily life for one week were measured using JD Mate. These parameters were collected as pre-IWT data.

On the day of starting IWT, the participants underwent a medical examination; blood samples were collected for biochemical analysis.

During the study, the participants visited the hospital and were examined at 4-week intervals. Training records obtained using JD Mate were transferred to the central server computer via the Internet. Each participant received an evaluation of the previous 2-week training data at each visit, encouraging them to continue IWT. The aforementioned remote individual exercise prescription system was also developed by the Department of Sports of Medical Sciences, Shinshu University Graduate School of Medicine [14]. The doctor conducted a questionnaire survey regarding the presence and degree of trauma, upper and lower limb joint pain, and back pain from the start to each visit.

In the 18th week of the study, the participants wore a FGM device for two weeks and underwent a dietary survey using the BDHQ. Moreover, we measured the calories burned in daily life using JD Mate for one week. These parameters were collected as post-IWT data.

At the end of the intervention, weight, height, blood pressure, and biochemical data were recorded.

Within two months of the end of the study period, the participants underwent $VO_2$ peak measurement, muscle strength measurement, abdominal and thigh plain CT, and plain abdominal MRI as before the study. The participants continued the IWT until these examinations were fully completed.

## Biochemical data

Venous blood samples were collected on the first and last visits. HbA1c levels were determined using high-performance liquid chromatography, while blood C-peptide levels and IRI were measured using an enzyme immunoassay (Tosoh Corp., Tokyo, Japan). HDL and LDL cholesterol levels were measured using the homogenous method (Sekisui Medical Corp., Tokyo, Japan). Triglyceride levels were measured using the glycerol elimination method. BUN levels were measured using a urease UV method kit (Quick-Auto Neo UN) (Shino-Test Corp.,

Tokyo, Japan). Serum creatinine levels were enzymatically measured (Sinotest Corp., Tokyo, Japan). Urine albumin levels were measured using immunoturbidity (Fuji Film Corp., Osaka, Japan). Blood glucose, blood C-peptide, IRI, and triglyceride levels were measured at any time, not requiring fasting.

## MAGE measurement

An FGM device (FreeStyle Libre Pro) was worn for two weeks at the beginning of the study and two weeks from the 18th week of the study. MAGE [20], an index of the average blood glucose fluctuation exceeding one standard deviation, was calculated from the data.

## Image inspection

The MRI-derived proton density fat fraction (MRI-PDFF) measured liver fat content. To eliminate the effects of cardiac and intestinal peristaltic motion artifact, we placed two regions of interest in the peripheral areas of the right anterior lobe and the right posterior lobe of the liver [21]. Moreover, the average values were compared before and after the intervention. Compared with liver biopsy, MRI-PDFF measurement is stable and highly reproducible [22, 23]. Visceral and subcutaneous fat volume was measured at the level of the umbilicus line using plain CT [24]; furthermore, muscle mass was measured using plain CT at the center of the right thigh [25].

## Muscle strength measurement

We measured the extension and flexion force of both knees using a dynamometer and calculated the average value for each knee.

## Physical fitness test

Maximum physical fitness ($VO_2$ peak) was measured using a JD mate. The participants were asked to gather at the gymnasium and wear the JD mate on their waists. With encouragement from a JTRC trainer, the participants walked at gradually increasing speeds (3 graded subjective velocities: low, medium, and high speed; 3 minutes for each speed) and finally at the fastest walking speeds. Within 30s of the last fastest walking, energy consumption was defined as the maximum physical strength ($VO_2$ peak). The maximal oxygen uptake (liter/min, y) determined using this method has good agreement with the value (x) measured using a bicycle ergometer ($y = 1.1x - 0.16$, $r = 0.91$, $p < 0.0001$) [13].

## Daily calorie intake and consumption survey

The BDHQ [19], a questionnaire that estimates energy intake by asking consumption details for 58 foods and beverages, was used to conduct dietary surveys to estimate dietary calorie intake. This dietary survey method has little seasonal variation in estimating energy intake [26]. Daily calorie consumption was calculated using JD Mate. The time-weighted average of energy expenditure by daily basal physical activity was calculated by adding all numbers derived by the following formula: (each measurement time / total measurement time) × calorie burned corresponding to each measurement time. Statistical significance was set at $p < 0.05$.

## Sample size

Since this study is a prospective, single-arm pilot study, this study aims to demonstrate descriptive statistics of IWT adherence and changes in various data before and after the intervention of IWT. Therefore, we did not calculate the sample size setting by power analysis, as

our primary goal was not to perform hypothesis testing. Regarding feasibility, 70 participants were recruited, and 51 were finally registered.

## Statistical analyses

Baseline clinical characteristics are expressed as the mean with standard deviation and percentages. Continuous variables are expressed as the mean with standard deviation. Changes in outcome variables after 20 weeks were assessed using a paired t-test and described as differences (pre IWT–post IWT) with 95% confidence intervals (CI). Since the number of samples was sufficiently large (n = 51), the t-test was used due to the central limit theorem [27]. A p-value of < 0.05 was considered statistically significant. Statistical analyses were performed using the statistical software R version 4.0.2 (2020-06-22) (R Foundation for Statistical Computing, Vienna, Austria). The effect size was calculated using *Cohen's d*. An effect size of $>|0.2|$ is considered small, $>|0.5|$ is medium, and $>|0.8|$ is large [28].

## Results

### Baseline characteristics of the participants

Table 1 shows the participant characteristics. We included 51 adults with T2D (29 persons, 56.9% men). The mean age and BMI were 62.3 ± 11.5 years and 27.1 ± 3.7 kg/m$^2$, respectively.

**Table 1. Baseline clinical characteristics of all participants.**

| | |
|---|---|
| Age, mean (SD) | 63.0 (11.5) |
| BMI, mean (SD) | 27.1 (3.7) |
| Male, n (%) | 29 (56.9) |
| Alcohol drinking, n (%) [a] | 35 (68.8) |
| Smoking, n (%) [b] | 2 (3.9) |
| Daily exercise habits, n (%) [c] | 11 (22.0) |
| Diabetes family history, n (%) [d] | 35 (68.6) |
| Hypertension, n (%) | 35 (68.6) |
| Dyslipidemia, n (%) | 35 (68.6) |
| Treatment: Drug usage rate, n (%) | |
| Insulin injection | 14 (27.5) |
| Metformin | 32 (62.7) |
| Sulfonylurea | 18 (35.3) |
| DPP4inhibitor | 27 (52.9) |
| SGLT2 inhibitor | 21 (41.2) |
| GLP-1 analogue | 14 (27.5) |
| Alpha-glucosidase inhibitor | 10 (19.6) |
| Glinide | 4 (7.8) |
| Pioglitazone | 1 (2.0) |
| Nephropathy, n (%) | |
| U-Alb < 30 mg/gCre | 34 (66.7) |
| U-Alb ≥ 30 mg/gCre | 17 (33.3) |

[a]: Including occasional drinking

[b]: Including past smoking history

[c]: Exercise for ≥ 30 minutes at least twice a week and continued for ≥ 1 year

[d]: Presence of diabetic relatives within the second degree.

BMI, body mass index; DPP4, dipeptidyl peptidase-4; SGLT2, sodium-glucose cotransporter 2: U-Alb, urine albumin; Cre, creatinine.

Only 22% of the participants had regular exercise habits; further, 68.6% of participants had hypertension, and a similar proportion had dyslipidemia. Additionally, 66.7% of the participants had urinary albumin excretion < 30 mg/g of creatinine. Metformin showed the highest usage rate (62.7%) among oral hypoglycemic agents, followed by dipeptidyl peptidase-4 inhibitors, sodium-glucose cotransporter 2 inhibitors, sulfonylureas, and insulin.

## Adherence to fast walking time

Fig 3 shows distribution of the fast-walking time and the number of people. The median total fast walking time was 1,022 minutes. Since the weekly target was 60 minutes of fast walking [10], participants with fast walking times > 1,200 minutes at 20 weeks (n = 20 [39%]; 12 men and eight women) were classified as the achievement group. Nevertheless, all 51 participants completed IWT for five months to the extent of their abilities.

## Glucose, HbA1c, and MAGE

The primary outcome results, differences in HbA1c before and after IWT, are shown in Table 2. The mean HbA1c levels before and after IWT were 7.33% and 7.45% (57 mmol/mol and 58 mmol/mol), respectively, which indicated no significant post-intervention changes in HbA1c levels. Additionally, there were no significant differences in blood C-peptide and IRI levels, and mean glucose levels were rather significantly increased (Table 2). Differences in the pre-and-post IWT mean values of MAGE with 95% confidence intervals were measured using an FGM device. No post-intervention change was observed in MAGE. Effect sizes for these data did not show significant improvement (Table 2). There was a post-intervention decrease and increase in the dose of diabetes drugs in 11 and eight people, respectively.

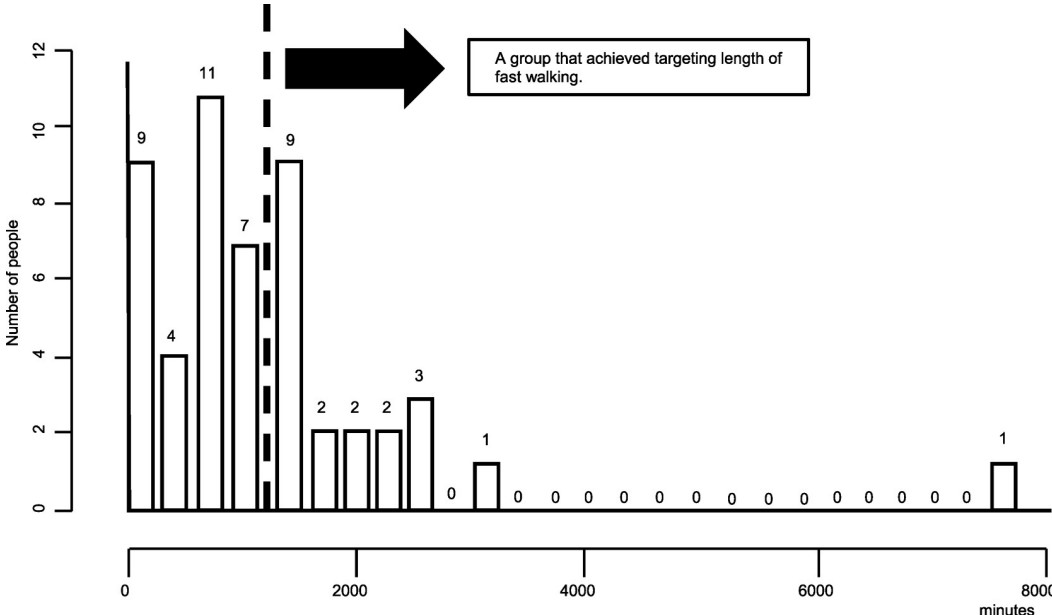

**Fig 3. Distribution of the fast-walking time and the number of people.** Each participant's total fast walking time was divided into sections of 300 min, with the number of participants in each section being shown. The target fast walking time was > 60 minutes per week; accordingly, participants who achieved > 1,200 minutes in fast walking time during the total study period (20 weeks) were defined as the achievement group.

**Table 2. Laboratory tests for glucose metabolism before and after IWT.**

|  | Pre IWT | Post IWT | Difference (pre—post) | 95% CI | p-value | Cohen's d |
|---|---|---|---|---|---|---|
| HbA1c (%), mean (SD) | 7.33 (0.81) | 7.45 (0.81) | -0.12 | [-0.28; 0.40] | 0.14 | 0.15 |
| Serum blood glucose (mg/dL), mean (SD) | 136.30 (45.80) | 153.50 (44.20) | -17.20 | [-31.60; -2.80] | 0.020 * | 0.38[†] |
| Serum C-peptide (ng/mL), mean (SD) | 3.37 (2.10) | 3.47 (2.22) | -0.10 | [-0.64; 0.44] | 0.71 | 0.05 |
| IRI (µU/mL), mean (SD) | 41.68 (65.85) | 41.06 (58.70) | 0.63 | [-8.23; 9.48] | 0.89 | -0.01 |
| MAGE, mean (SD) | 84.36 (24.82) | 85.09 (22.72) | -0.72 | [-5.11; 3.67] | 0.74 | 0.03 |

MAGE indicates the average blood glucose fluctuation exceeding 1 standard deviation. The MAGE values are based on calculating the glucose profiles recorded for each person over 14 days using FreeStyle Libre Pro.

*p < 0.05

[†] Cohen's d > |0.20|

IWT, interval walking training; HbA1c, hemoglobin A1c; IRI, immunoreactive insulin; MAGE, mean amplitude of glycemic excursions.

## Body composition by CT and MRI

Table 3 indicates the data of the secondary outcome, changes in the body composition before and after IWT. One participant could not undergo MRI because he had a pacemaker. In both statistical hypothesis testing and effect size analysis, liver fat mass (PDFF), visceral fat mass, subcutaneous fat mass, and thigh muscle mass showed no significant post-intervention improvements before and after IWT. However, PDFF showed a post-intervention decreasing trend (8.73% before IWT, 7.59% after IWT, $p = 0.051$), and similarly, there was a post-intervention decreasing trend in visceral fat (183.58 cm$^2$ before IWT and 174.15 cm$^2$ after IWT, $p = 0.065$).

## Biochemical data

Table 4 shows the biochemical data and BMI before and after IWT. There was a significant post-intervention increase in HDL cholesterol (from 1.4 mmol/L to 1.5 mmol/L, $p = 0.0093$, *Cohen's d* = 0.25) and BUN levels ($p = 0.047$, *Cohen's d* = 0.23). In effect size analysis, triglyceride levels showed significant improvement, and urine albumin levels showed significant deterioration (*Cohen's d* = -0.55, 0.74, respectively). There were no significant changes in the other values.

## Physical fitness and muscle mass strength

Table 5 shows VO$_2$ peak and thigh muscle strength values before and after IWT. As for muscle strength, there was an upward trend in extension muscle strength (430.1 N before IWT and 448.4 N after IWT, $p = 0.064$), but no significant difference was observed. The target fast walking time was set to $\geq$ 60 minutes per week due to the report that the effect of increasing VO$_2$

**Table 3. Body composition data obtained using CT and MRI before and after IWT.**

|  | Pre IWT | Post IWT | Difference (pre—post) | 95% CI | p-value | Cohen's d |
|---|---|---|---|---|---|---|
| Liver fat mass (PDFF: %) [a], mean (SD) | 8.73 (6.86) | 7.59 (5.74) | 1.14 | [-0.003; 2.28] | 0.051 | -0.17 |
| Visceral fat mass (cm$^2$), mean (SD) | 183.58 (77.22) | 174.15 (81.35) | 9.43 | [-0.61; 19.47] | 0.065 | -0.12 |
| Subcutaneous fat mass (cm$^2$), mean (SD) | 158.76 (72.11) | 152.85 (65.26) | 5.90 | [-1.44; 13.25] | 0.11 | -0.08 |
| Thigh muscle mass (cm$^2$), mean (SD) | 118.90 (23.12) | 117.58 (21.60) | 1.32 | [-0.95; 3.59] | 0.25 | -0.06 |

a: anterior and posterior segment.

CT, computed tomography; MRI, magnetic resonance imaging; IWT, interval walking training; PDFF, proton density fat fraction.

**Table 4. Laboratory tests before and after IWT.**

| | Pre IWT | Post IWT | Difference (pre—post) | 95% CI | p-value | Cohen's d |
|---|---|---|---|---|---|---|
| HDL-C, (mmol/L), mean (SD) | 1.4 (0.4) | 1.5 (0.4) | -0.07 | [-0.1; -0.02] | 0.0093** | 0.25[†] |
| LDL-C, (mmol/L), mean (SD) | 2.7 (0.6) | 2.7 (0.6) | -0.06 | [-0.2; 0.08] | 0.40 | 0.00 |
| Triglyceride, (mg/dL), mean (SD) | 1.8 (1.1) | 1.2 (0.7) | 0.2 | [-0.02; 0.4] | 0.075 | -0.55[†] |
| BUN, (mmol/L), mean (SD) | 5.6 (1.3) | 5.9 (1.4) | -0.3 | [-0.6; -0.03] | 0.047* | 0.23[†] |
| Cre, (umol/L), mean (SD) | 72.5 (17.7) | 72.5 (16.8) | -0.9 | [-2.7; 0.9] | 0.46 | 0.00 |
| AST, (IU/L), mean (SD) | 26.8 (19.7) | 25.1 (14.5) | 1.0 | [-1.8; 3.8] | 0.48 | -0.09 |
| ALT, (IU/L), mean (SD) | 32.0 (29.8) | 30.6 (31.1) | 0.2 | [-3.2; 3.7] | 0.90 | -0.05 |
| eGFR, (mL/min/1.73m$^2$), mean (SD) | 69.6 (17.4) | 67.9 (14.8) | 1.7 | [-0.4; 3.7] | 0.11 | -0.10 |
| Urine albumin, (mg/g·Cre), mean(SD) | 38.6 (60.9) | 83.9 (253.0) | -44.0 | [-102.3; 14.4] | 0.14 | 0.74[†] |
| BMI, (kg/m$^2$), mean (SD) | 27.1 (3.8) | 26.9 (3.6) | 0.2 | [-0.03; 0.5] | 0.084 | -0.05 |
| **Blood pressure** | | | | | | |
| systolic, (mmHg), mean (SD) | 124.5 (14.3) | 122.3 (11.8) | 1.2 | [-3.0; 5.4] | 0.56 | -0.15 |
| diastolic, (mmHg), mean (SD) | 75.9 (12.2) | 76.1 (9.6) | 0.2 | [-3.4; 3.9] | 0.90 | 0.02 |

*p < 0.05

**p < 0.005

[†] Cohen's d > |0.20|

IWT, interval walking training; HDL-C, high-density lipoprotein; LDL-C, low-density lipoprotein; BUN, blood urea nitrogen; Cre, creatinine; AST, aspartate aminotransferase; ALT, alanine aminotransferase; eGFR, estimated glomerular filtration rate; BMI, body mass index.

peak almost plateaued at 50 minutes per week [10]. Therefore, in the analysis of the achievement group only, the VO$_2$ peak was significantly increased by 10% after IWT (from 1,682 mL/min to 1,827 mL/min after, $p = 0.0374$, *Cohen's d* = 0.24), which was similar to conventional data of healthy people [12].

## Energy intake and consumption

No significant difference showed in the daily energy intake (2008 and 1943 kcal before and after IWT, respectively, $p = 0.34$). There was a post-intervention decrease in the lipid intake (from 68.1 g/day to 62.1 g/day, $p = 0.034$); however, no significant post-intervention change

**Table 5. Physical fitness and muscle strength before and after IWT.**

| | Pre IWT | Post IWT | Difference (pre—post) | 95% CI | p-value | Cohen's d |
|---|---|---|---|---|---|---|
| **VO$_2$ peak, (mL/min), mean (SD)** | | | | | | |
| all [a](n = 51) | 1,672 (673) | 1,705 (696) | -29 | [-112; 56] | 0.51 | 0.05 |
| fast walking time ≥1200 min [b](n = 20) | 1,682 (615) | 1,827 (650) | -147 | [-282; -10] | 0.037* | 0.24[†] |
| fast walking time <1200 min [c](n = 31) | 1,655 (716) | 1,618 (720) | 46 | [62; 153] | 0.39 | -0.05 |
| **Thigh muscle strength, (N), mean (SD)** | | | | | | |
| Extension muscle strength | 430.1 (151.8) | 448.4 (131.9) | -18.3 | [-37.6; 1.1] | 0.064 | 0.12 |
| Flexion muscle strength | 261.1 (71.5) | 254.3 (72.5) | 6.8 | [-4.6; 18.2] | 0.24 | -0.10 |

*p < 0.05

[†] Cohen's d > |0.20|

[a]: all participants in the study

[b]: participants who reached a fast walking time of ≥1200 min.

[c]: participants with a fast walking time of <1200 min.

IWT, interval walking training.

was revealed in intake of carbohydrate, protein, dietary fiber, and salt. The daily basal physical activity energy expenditure, excluding calories burned during IWT, and calculation at the weighted average, which was calculated from JD mate, was 108.1 kcal and 101.5 kcal before and after IWT, respectively. Their energy expenditure indicated a non-significant post-intervention decrease ($p = 0.054$).

### Adverse events

Regarding locomotor disorders resulting from IWT, the participants completed a questionnaire at each visit regarding the presence and degree of "joint pain, low back pain, and trauma" during the IWT period. Six persons (12%) complained of mild exacerbation of joint pain due to IWT; however, none of the participants wished to discontinue IWT due to these factors. Severe hypoglycemia was not observed during IWT partly because of the adjustment of diabetic drugs.

## Discussion

We demonstrated descriptive statistics of IWT adherence and changes in various data before and after the intervention of IWT in adults with T2D. Further, statistical hypothesis testing and effect size calculation were performed in this study. Although only 39% of the participants achieved the target period of fast walking, all included participants performed the IWT for five months. This study had no significant post-intervention improvements in HbA1c and blood glucose-related data. However, regardless of the amount of fast walking time, we observed a significant increase in HDL cholesterol levels, an improvement tendency in triglycerides, BMI, hepatic fat mass, abdominal visceral fat mass, and muscle strength. In addition, the participants who achieved the target fast walking time showed a significant increase in $VO_2$ peak. Previously, physical fitness could not be increased without applying a high workload. Contrastingly, we found that IWT, an easy-to-perform and available exercise method, could successfully improve physical fitness if the target time of fast walking was achieved. Given the extent of changes in the aforementioned parameters, it is necessary to consider energy intake and consumption. However, there were no significant changes in dietary intake and energy consumption in daily life before and after the intervention. Therefore, the improvements above could be considered almost wholly attributed to the effect of IWT.

It should be noted that although the changes in HDL-C in the whole group and $VO_2$ peak in the target achievement group are statistically significant, the effect sizes are small, *Cohen's d* = 0.25, 0.24, respectively, and, therefore, cannot be considered clinically fully significant. Contrastingly, for changes in triglyceride levels of which the *p*-value was not significant, the effect size was considered intermediate with *Cohen's d* = -0.55. These effect sizes should be used for future RCT studies to design appropriate sample sizes, and the detailed effects of IWT, focusing on these parameters, will be examined.

There are some considerations regarding the lack of improvement in blood glucose-related data. The possible causes are as follows. First, there were changes in diabetes drug dosage. Due to the relatively large number of insulin (27.5%) and sulfonylureas (35.3%) users in this study, drug changes were freely permitted to avoid hypoglycemia caused by IWT. As a result, the doses were reduced in 11 (22%) individuals, making it difficult to assess changes in HbA1c. Second, there were seasonal fluctuations in glycemic control for half a year. This study started in July-October and ended in February-April, thus including observing changes in HbA1c from summer to winter. HbA1c has been reported [29] to be 0.22% higher in winter (January to April) than in summer (July to October), and this tendency was observed in low temperatures ($< 0°C$) during winter. In the Nagano prefecture, where this study was conducted, the

temperature often falls below 0°C in winter; therefore, changes in HbA1c may have been affected by seasonal fluctuations.

Exercise therapy is a fundamental treatment for T2D. Generally, it is recommended that adults with T2D perform a moderate-to-intense aerobic exercise (maximum oxygen uptake $\geq$ 50%) for 150 minutes a week and avoid $>$ two consecutive days of no exercise [6]. However, given the vagueness of this standard, it is difficult to provide specific guidelines in the hospital's outpatient department, which has led to a decrease in the rate of continuation of exercise in persons with T2D. In this study, we provided raw data regarding the continuation of IWT in the outpatient department of diabetes and showed that the continuation rate of IWT was high. Previous studies only targeted participants who achieved their target momentum. In reality, many outpatients cannot reach the target exercise amount; moreover, data only from some "excellent" participants with diabetes cannot reflect the versatility of specific exercise therapy. Accordingly, in this study, 20 (39%) participants reached the target exercise amount (Fig 2). However, all participants continued IWT to the end within the allowed range (continuation rate: 100%), which indicated the high versatility of IWT. Moreover, locomotor disorders due to IWT were all mild and could not cause discontinuation of IWT. These adherence and safety data suggest the high versatility of IWT according to each individual's ability.

IWT is based on the concept that exercise training at an intensity above the anaerobic range value effectively increases aerobic capacity. However, such exercise cannot be performed over long periods since it creates acidosis due to lactic acid accumulation. Accordingly, a light exercise interval is placed to allow the body to recover from acidosis. Regarding the strength of $\geq$ 70% of the $VO_2$ peak in fast walking, we referred to the American College of Sports Medicine's recommendation of exercise intensity of $\geq$ 60% of the maximal oxygen uptake for improving physical fitness [30]. Additionally, we chose a strength of 70% of the $VO_2$ peak based on the experience that most middle-aged and older people could walk at high speeds with 70% intensity of the $VO_2$ peak in our center. In this study, the most characteristic feature of IWT is that participants in the achievement group (fast walking for $\geq$ 60 minutes per week: $\geq$ 1200 minutes for 20 weeks) showed an average increase in physical fitness of 10%, which is similar to the results of healthy people. Given the presence of sarcopenia and frailty in persons with T2D, it is essential to increase their physical fitness; moreover, doctors applying IWT to individuals with T2D should be aware of this benefit. Previous studies have reported that for 5-month exercise therapy to increase physical fitness in persons with T2D, exercise should be performed at an intensity of 50–75% of the $VO_2$ peak for approximately 50 minutes per day and 3–4 times a week [31]. This exercise amount increased the $VO_2$ peak by 11.8%. Contrastingly, IWT could improve physical fitness with less exercise, suggesting that IWT can increase physical fitness more effectively.

There are other studies on IWT applied to T2D. A Denmark-based group has developed the "InterWalk application," which was inspired and arranged by the JD Mate [32]. IWT using the InterWalk app has been reported to improve glycemic control and exercise adherence for individuals with T2D [32, 33]. However, the InterWalk app is different from JD Mate in that fast and slow walking intensity is self-selective (subjective) and is not determined by the $VO_2$ peak. Therefore, it is not easy to make a simple comparison between their results and ours.

This study had some limitations. First, 39% of the included participants achieved the target exercise amount, which was a lower percentage than expected. In studies on conventional IWT, the achievement rate was approximately 60–90% [17, 18, 34]. However, these studies included many of healthy people who participated in their interests. Additionally, they received frequent advice from JTRC trainers. Second, levels of blood glucose, blood C-peptide, IRI, and triglyceride were measured at any time, not requiring fasting, because some of the participants had difficulty coming to the hospital early in the morning. However, blood

samples were collected at about the same time on the first and last visit. Third, since this was a single-group intervention study of the utility of IWT in conventional practice, our findings cannot be strictly attributed to the effect of IWT. To compensate for a weakness of the single-arm study as much as possible, we examined changes in dietary intake and daily life energy consumption before and after the study, and no significant changes were found. This study is useful in estimating the appropriate sample size for designing a future comparative study. Therefore, based on the results of this study alone, we do not describe the effects of IWT completely but rather regard it as a preliminary study for subsequent comparative studies.

In this prospective pilot study, we examined the applicability of IWT in Japanese individuals with T2D. All 51 participants completed the IWT for five months; furthermore, in terms of effect sizes, IWT was suggested to have a positive effect on lipid metabolism and physical fitness. In future RCT studies, the detailed effects of IWT, focusing on these parameters, will be examined. IWT can allow the achievement of physical fitness that could only be previously achieved by applying a high load at the gym. Accordingly, it could be said that IWT is an exercise therapy that combines the effects of both aerobic and resistance exercises. Therefore, IWT has great potential to spread worldwide as exercise therapy for individuals with T2D.

## Supporting information

**S1 Checklist. TREND statement checklist.**
(PDF)

**S1 File.**
(DOCX)

**S2 File.**
(DOCX)

## Acknowledgments

This work is supported by the Center for Clinical Research, Shinshu University Hospital. We wish to thank Prof. Masayoshi Koinuma (Center for Clinical Research, Shinshu University Hospital and Faculty of Pharmaceutical Sciences, Teikyo Heisei University) for the helpful discussions. We wish to thank Prof. Takeji Umemura (Division of Gastroenterology, Department of Internal Medicine, Shinshu University School of Medicine) for the valuable advice on the study design. We want to thank Editage (www.editage.com) for English language editing and SRD Holdings Co., Ltd. (www.srd-hd.co.jp) for statistical analyses.

## Author Contributions

**Data curation:** Takahiro Miyakoshi, Manami Hosokawa, Mayuka Furihata, Yosuke Okubo, Ai Sato, Masanori Yamazaki.

**Formal analysis:** Masaaki Takahashi.

**Methodology:** Shizue Masuki, Hiroshi Nose.

**Project administration:** Ako Oiwa, Takahiro Miyakoshi.

**Supervision:** Mitsuhisa Komatsu.

**Writing – original draft:** Kouhei Kitajima, Ako Oiwa.

**Writing – review & editing:** Masanori Yamazaki, Mitsuhisa Komatsu.

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
