## [Decision Letter · Decision Letter 0]

28 Feb 2023

PONE-D-22-23027Interval walking training in type 2 diabetes: A pilot study to evaluate the applicability as exercise therapyPLOS ONE

Dear Dr. Oiwa,

Thank you for submitting your manuscript to PLOS ONE. After careful consideration, we feel that it has merit but does not fully meet PLOS ONE’s publication criteria as it currently stands. Therefore, we invite you to submit a revised version of the manuscript that addresses the points raised during the review process.

We look forward to receiving your revised manuscript.

Kind regards,

Jennifer Annette Campbell, PhD, MPH

Academic Editor

PLOS ONE

Journal Requirements:

Reviewers' comments:

Reviewer's Responses to Questions

**Comments to the Author**

1. Is the manuscript technically sound, and do the data support the conclusions?

Reviewer #1: Yes

Reviewer #2: Partly

Reviewer #3: Yes

2. Has the statistical analysis been performed appropriately and rigorously? 

Reviewer #1: Yes

Reviewer #2: Yes

Reviewer #3: Yes

3. Have the authors made all data underlying the findings in their manuscript fully available?

Reviewer #1: Yes

Reviewer #2: Yes

Reviewer #3: No

4. Is the manuscript presented in an intelligible fashion and written in standard English?

Reviewer #1: No

Reviewer #2: No

Reviewer #3: Yes

5. Review Comments to the Author

Reviewer #1: A single-arm intervention clinical pilot trial was conducted which aimed to evaluate the efficacy of interval walking training (IWT) on glucose and lipid metabolism, body composition, physical fitness, and muscle strength in subjects with type 2 diabetes. No statistically significant changes in the primary outcomes (HbA1c and body composition) were observed. The target achievement subgroup showed a statistically significant increase in VO2 peak.

Minor revisions:

1- Abstract: Clarify which outcomes are primary and which are secondary.

2- Abstract: List the statistical methods used for estimating the p-values.

3- Remove the results where a trend is claimed.

4- Indicate if the distribution of the data was checked for normality prior to applying paired t-tests.

5- Line 252: Include the standard deviations that correspond to the means.

6- It is standard practice to summarize normally distributed data using means and standard deviations. For non-normally distributed data, use median, first and third quartiles.

7- Thoroughly proofread the manuscript. Sometimes the phraseology is non-standard.

Reviewer #2: General Comments:

In this paper the authors presents a study to evaluate the effect of interval walking training on glucose and lipid metabolism, body composition, physical fitness and muscle strength in persons with type 2 diabetes, which sounds novel and original since many studies have not been published in this area, but at the moment the manuscript has not been presented in an intelligible fashion and the method has not been described in sufficient detail. There are grammar and language issues across the manuscript that require attention and proof reading. Please thoroughly check it.

Abstract

In the materials and methods (page 2 line 34-38) the frequency and intensity of the intervention should be clearly stated.

In the results (page 3 line 39-51), no significant effects should be stated as ‘though no significant, there tend to be improvement in ……’

In the conclusion (page 3 line 52-55), with no improvement in the main study outcome and only one secondary outcome, this section should be written modestly in accordance with the results.

Introduction

The introduction does not provide sufficient background and do not include all relevant references. The sections needs to be rewritten for clarity of purpose and to reflect novelty, rigour and impact as well as rationale for this study.

Page 4 line 63: add ‘combined’ aerobic and resistance exercise...

Page 4 line 64: change ‘effectively’ to ‘effective’

Page 4 line 67: change ‘side’ to ‘hand’, rephrase to ‘there is a low uptake of exercise, with…’

Page 4 line 69: remove ‘people’

Page 4 line 70-71: rephrase to ‘Another reason for the low exercise uptake among people with type 2 diabetes could be difficulty on the part of medical professionals to teach ….’

Page 5-6 line 73-101: this should be summarised to provide a rational for the current study. The details of the protocol should only be referenced. Example ‘details of the protocol is published elsewhere (aa et al., 1999)’

Materials and methods

This is not adequately described and must be improved significantly. Currently, it is difficult to replicate in terms of the description provided.

Research design

Page 8 lines 124-126: Unclear

Line 128: Participant registration should be ‘Participants’ only

Page 9 line 129: How were participants recruited? Why was BMI 20-34kg/m2 an inclusion criterion?

Intervention protocol

The outcomes and their respective measures, how, when and where they were taken should be clearly stated, and should be delineated under the section ‘Outcome measures’ before the 'Intervention protocol'.

The intervention should be clearly outlined. Currently it is difficulty to follow. Would be helpful if it done chronologically.

Page line 194-198: MAGE measurement

Did participants wear the device throughout the 20weeks of the study? It is mentioned here 2weeks before start of study and for 18 weeks after the start of the study. In the intervention it was mentioned that 18th week of the study. Please reconcile.

Line 224: What is fastest walking? How was it determined? How is low, medium and high speed differentiated?

Results

Needs to be rewritten for clarity. May be better to state clearly significant findings followed by non-significant findings or vice-versa under each section, rather than mix them together

Line 250: Change to ‘Basal’ to ‘Baseline’ characteristics

Page 15 line 254: ‘hypertension and dsylipdemia showed 68.6% is unclear.

Line 328-331: Please state non-significant finding as such. The target fast walking time needs to be properly explained from the onset in the methods.

Line 334: reference needed.

Line 335: Table 5. What is all (51) and the associated values?

Discussion

This section should be written to provide a more holistic perspective of the current findings in relation to literature.

Reviewer #3: Introduction

In line 68-70, the authors stated without appropriate citation, “This low practice rate could be attributed to low people motivation, limited time availability, management problems, and lack of willpower or control.”

In line 73-76, the authors stated without appropriate citation, “The Department of Sports Medical Sciences, Shinshu University Graduate School of Medicine, developed a unique walking method termed interval walking training (IWT) in 1999. There has been active research and development of IWT, with approximately 8,700 individuals of overall good health participating in these studies.”

In line 92-93, the authors only cited one article but stated, “There is a plethora of scientific evidence supporting the efficacy of IWT, with increasing worldwide attention [11].”

Materials and Methods:

Did the authors do any power calculation? How did the authors arrive at 51 participants as the appropriate sample size for the pilot study?

How did the authors arrived at 20 weeks of follow-up? Why not 6 months or a year?

Results:

In line 44-49, the authors stated, “However, there was significant improvement in high density lipoprotein cholesterol (from 1.4 mmol/L to 1.5 mmol/L, p = 0.0093). Further, there was improvement trend in liver fat mass (from 8.73% to 7.59%, p = 0.051), visceral fat mass (from 183.58 cm2 to 174.15 cm2, p = 0.065), and extension muscle strength (from 430.1 N to 448.4 N, p = 0.064). In the target achievement group, a significant increase in VO2 peak by 10% (from 1,682 mL/min to 1,827 mL/min, p = 0.037) was observed.” The changes in liver fat mass, visceral fat mass, and extension muscle strength are not statistically significant based on their p-values. Why did the authors state otherwise?

The change in HDL (from 1.4 mmol/L to 1.5 mmol/L, p = 0.0093) is statistically significant, does the authors think the change in HDL is clinically significant?

6. PLOS authors have the option to publish the peer review history of their article (what does this mean?). If published, this will include your full peer review and any attached files.

Reviewer #1: No

Reviewer #2: No

Reviewer #3: No

---

## [Author Response · Author response to Decision Letter 0]

12 Apr 2023

Dear Reviewer 1,

We are grateful to Reviewer 1 for the constructive comments and useful suggestions that have helped us to improve our paper considerably. As indicated in the responses that follow, we have taken all these comments and suggestions into account in the revised version of our paper.

Comment #1- Abstract: Clarify which outcomes are primary and which are secondary.

Response: We have clearly stated the primary and secondary outcomes in the Abstract section. (lines 46-47)

Comment #2- Abstract: List the statistical methods used for estimating the p-values.

Response: We have added a description of the t-test to the Abstract section. (line 49 and line 51)

Comment #3- Remove the results where a trend is claimed.

Response: We have removed expressions such as “trend” and instead stated the results. (lines 46-51)

Comment #4- Indicate if the distribution of the data was checked for normality prior to applying paired t-tests.

Response: We thank the reviewer for the valuable comment. Since the number of samples is sufficiently large (n = 51), due to the idea of the central limit theorem, we applied a paired t-test without checking for normality. We have added this to statistical analyses in the Materials and Methods section. (lines 272-273)

Comment #5- Line 252: Include the standard deviations that correspond to the means.

Response: We thank the reviewer for pointing this out. We have added standard deviations where appropriate. (line 282)

Comment #6- It is standard practice to summarize normally distributed data using means and standard deviations. For non-normally distributed data, use median, first and third quartiles.

Response: As the reviewer pointed out, we believe that it is a fundamental and very important issue from a statistical point of view. We consulted statistics experts at the study planning stage. As described in lines 272-273 of the Materials and Methods section, the number of participants in this study was sufficiently large (n = 51): therefore, based on the concept of the central limit theorem, we judged that the data of this study could be statistically regarded as a having normal distribution.

Comment #7- Thoroughly proofread the manuscript. Sometimes the phraseology is non-standard.

Response: We apologize for the inappropriate expressions in some of the sentences. We have thoroughly proofread the manuscript. The manuscript has undergone professional English proofreading again.

Yours sincerely, 

Ako Oiwa MD., Ph.D.

Dear Reviewer 2,

We are grateful to Reviewer 2 for the constructive comments and useful suggestions that have helped us to improve our paper considerably. As indicated in the responses that follow, we have taken all these comments and suggestions into account in the revised version of our paper.

Abstract:

Comment #1

In the materials and methods (page 2 line 34-38) the frequency and intensity of the intervention should be clearly stated.

Response: We thank the reviewer for pointing this out. As suggested by the reviewer, we have added these details to the Abstract section. (lines 37-44)

Comment #2

In the results (page 3 line 39-51), no significant effects should be stated as ‘though no significant, there tend to be improvement in ……’

Response: As per the reviewer’s comment, we have corrected the Abstract section for improved clarity. We have removed the word "tend or trend." (lines 45-55)

Comment #3

In the conclusion (page 3 line 52-55), with no improvement in the main study outcome and only one secondary outcome, this section should be written modestly in accordance with the results.

Response: We agree with the reviewer’s view. We have revised the Abstract section. As pointed out by other reviewers, we added effect size calculations and rewrote the Abstract accordingly. (lines 55-58)

Introduction

The introduction does not provide sufficient background and do not include all relevant references. The sections needs to be rewritten for clarity of purpose and to reflect novelty, rigour and impact as well as rationale for this study.

Comment #4: Page 4 line 63: add ‘combined’ aerobic and resistance exercise...

Response: We thank the reviewer for pointing this out. We have corrected the relevant text. (line 66)

Comment #5: Page 4 line 64: change ‘effectively’ to ‘effective’

Response: We thank the reviewer for pointing this out. We have corrected the relevant text. (line 67)

Comment #6: Page 4 line 67: change ‘side’ to ‘hand’, rephrase to ‘there is a low uptake of exercise, with…’

Response: We thank the reviewer for pointing this out. We have corrected the relevant text. (line 70) 

Comment #7: Page 4 line 69: remove ‘people’

Response: We have corrected the relevant text. (line 72)

Comment #8: Page 4 line 70-71: rephrase to ‘Another reason for the low exercise uptake among people with type 2 diabetes could be difficulty on the part of medical professionals to teach ….’

Response: We have corrected the relevant text. (lines 73-74)

Comments #9: Page 5-6 line 73-101: this should be summarised to provide a rational for the current study. The details of the protocol should only be referenced. Example ‘details of the protocol is published elsewhere (aa et al., 1999)’

Response: We thank the reviewer for pointing this out. We have corrected the relevant text, provided a clearer summary, and provided references for further study details. (lines 76-99)

Material and Methods

Research design

Comments #10: Page 8 lines 124-126: Unclear

Response: As the reviewer pointed out, the explanation of Fig.2 was omitted. We have added the description of Fig.2 in detail. (lines 133-142)

Comments #11: Line 128: Participant registration should be ‘Participants’ only.

Response: We have corrected the relevant text. (line 147)

Comments #12: Page 9 line 129: How were participants recruited? Why was BMI 20-34kg/m2 an inclusion criterion?

Response: We have added the method of recruitment. (lines 148-151) Since this study was a pilot study and adverse events due to IWT could not be predicted at the beginning of the study, lean persons or individuals with obesity were excluded to avoid the risk of the onset of locomotor disorders. The minimum average BMI for each age group of Japanese people is from 20 to 21 (women in their twenties): therefore, the minimum BMI for selection criteria is 20. The maximum BMI of 34 was slightly less than the definition for patients with obesity, 35. In other words, we used the BMI (20 to 34), which we empirically believe to be safe for Japanese patients with diabetes to exercise.

Intervention Protocol

Comments #13: The outcomes and their respective measures, how, when and where they were taken should be clearly stated, and should be delineated under the section ‘Outcome measures’ before the 'Intervention protocol'.

The intervention should be clearly outlined. Currently it is difficulty to follow. Would be helpful if it done chronologically.

Response: We thank the reviewer for the valuable comment. We agree with the reviewer that the original text made it difficult to understand our protocol. We have added a new section, “Outcome measures” before the Intervention protocol section. (lines 159-173) In addition, we have corrected the Intervention protocol so that it is in a chronological easy-to-understand manner. (lines 175-205)

Comment #14: Page line 194-198: MAGE measurement: Did participants wear the device throughout the 20weeks of the study? It is mentioned here 2weeks before start of study and for 18 weeks after the start of the study. In the intervention it was mentioned that 18th week of the study. Please reconcile.

Response: We thank the reviewer for the detailed comments. We have corrected the MAGE measurement in the Materials and methods section (line 221), as follows. “An FGM device (FreeStyle Libre Pro) was worn for two weeks at the beginning of the study and two weeks from the 18th week of the study.”

Comment #15: Line 224: What is fastest walking? How was it determined? How is low, medium and high speed differentiated?

Response: Low, medium, and high speed are subjective judgments made by participants. We have included this in line 243 of the revised manuscript.

Results

Needs to be rewritten for clarity. May be better to state clearly significant findings followed by non-significant findings or vice-versa under each section, rather than mix them together

Comments #16: Line 250: Change to ‘Basal’ to ‘Baseline’ characteristics

Response: We have corrected the relevant text. (line 280)

Comments #17: Page 15 line 254: ‘hypertension and dsylipdemia showed 68.6% is unclear.

Response: We have corrected this as indicated. (line 284)

Comments #18: Line 328-331: Please state non-significant finding as such. The target fast walking time needs to be properly explained from the onset in the methods.

Response: We have changed the expression as suggested (lines 362-364): we have changed to clearly stated non-significant findings as such. At the beginning of the Material and Methods section (lines 123-125), we have explained the target fast walking time.

Comments #19: Line 334: reference needed.

Response: We have added a reference. (line 369)

Comments #20: Line 335: Table 5. What is all (51) and the associated values?

Response: For clarity, we have annotated all (n = 51) and ≥1200 min (n = 20) and <1200 min (n = 31) in Table 5. (lines 373-375)

Discussion

This section should be written to provide a more holistic perspective of the current findings in relation to literature.

Response: From the point of view of the effect size, we have clearly stated that this study is essential for future RCT studies. (lines 416-423, lines 500-502)

Yours sincerely, 

Ako Oiwa MD., Ph.D.

Dear Reviewer 3,

We are grateful to Reviewer 3 for the constructive comments and useful suggestions that have helped us to improve our paper considerably. As indicated in the responses that follow, we have taken all these comments and suggestions into account in the revised version of our paper.

Introduction

Comments #1: In line 68-70, the authors stated without appropriate citation, “This low practice rate could be attributed to low people motivation, limited time availability, management problems, and lack of willpower or control.”

Response: We have added a reference. (line 73, reference No. 8)

Comments #2: In line 73-76, the authors stated without appropriate citation, “The Department of Sports Medical Sciences, Shinshu University Graduate School of Medicine, developed a unique walking method termed interval walking training (IWT) in 1999. There has been active research and development of IWT, with approximately 8,700 individuals of overall good health participating in these studies.”

Response: We have added a reference. (line 86, 88) Other reviewers pointed out that this section was difficult to understand: therefore, we have changed the structure of the entire sentence above. (lines 76-91)

Comments #3: In line 92-93, the authors only cited one article but stated, “There is a plethora of scientific evidence supporting the efficacy of IWT, with increasing worldwide attention [11].”

Response: We thank the reviewer for the comments. We have added some necessary references. (line 91)

Comments #4: Materials and Methods:

Did the authors do any power calculation? How did the authors arrive at 51 participants as the appropriate sample size for the pilot study?

Response: We thank the reviewer for pointing out this important part of the core of statistics. We have corrected sentences in Sample size in the Materials and Methods section (lines 261-265), as follows. “Since this study is a prospective, single-arm pilot study, the purpose of this pilot study is to demonstrate descriptive statistics of IWT adherence and changes in various data before and after the intervention of IWT in adults with type 2 diabetes. Therefore, we did not calculate the sample size setting by power analysis, as our primary goal was not to perform a hypothesis testing. With a view to feasibility, 70 participants were recruited, and 51 cases were finally registered.”

In this paper, many phrases indicate the evaluation of effectiveness: therefore, the aim in the Abstract section (lines 34-36) has been rewritten. 

Further, we have added the results of effect size (Cohen's d), as effect size calculations in this study are essential to determine the sample size for future RCT studies. Therefore, by inserting sentences related to effect size, we have added and corrected the followings: Table 2, 3, 4, 5, lines 319-320, 334, 349-350, 416-423.

Comments #5: 

How did the authors arrived at 20 weeks of follow-up? Why not 6 months or a year?

Response: 

We thank the reviewer for the valuable comments. The reason for setting 20 weeks is that we have found that physical fitness improvement with IWT occurs after 20 weeks of intervention (Nemoto et al. Mayo Clin Proc 2007), and that physical fitness increases by 10–15% in the first 20 weeks and then maintains physical fitness with continued IWT (Masuki et al. J Appl Physiol 2015). It is possible to set it to 6 months or 1 year, however, based on these research results, we set 20 weeks as one period when the effect of IWT can be recognized. We have added this in Research Design in the Materials and Methods section. (lines 125-128)

Comments #6: Results:

In line 44-49, the authors stated, “However, there was significant improvement in high density lipoprotein cholesterol (from 1.4 mmol/L to 1.5 mmol/L, p = 0.0093). Further, there was improvement trend in liver fat mass (from 8.73% to 7.59%, p = 0.051), visceral fat mass (from 183.58 cm2 to 174.15 cm2, p = 0.065), and extension muscle strength (from 430.1 N to 448.4 N, p = 0.064). In the target achievement group, a significant increase in VO2 peak by 10% (from 1,682 mL/min to 1,827 mL/min, p = 0.037) was observed.” The changes in liver fat mass, visceral fat mass, and extension muscle strength are not statistically significant based on their p-values. Why did the authors state otherwise?

Response: We agree with the reviewer’s point. Based on the view that only significant differences are statistically meaningful, we have changed the Result in the Abstract section. (lines 45-55)

Comments #7:

The change in HDL (from 1.4 mmol/L to 1.5 mmol/L, p = 0.0093) is statistically significant, does the authors think the change in HDL is clinically significant?

Response: We thank the reviewer for valuable comment. Regarding the amount of change in HDL-C, the effect size (Cohen’s d) is 0.25, which is a "small effect size": therefore, it is difficult to say that it is clinically fully significant. In addition, the VO2 peak in the target achievement group, which had a significant difference in the p-value, also had a "small effect size." In the Discussion section, we have added the sentence "The effect size was small." (lines 416-423)

Yours sincerely, 

Ako Oiwa MD., Ph.D.

---

## [Editor Report · Decision Letter 1]

2 May 2023

Interval walking training in type 2 diabetes: A pilot study to evaluate the applicability as exercise therapy

PONE-D-22-23027R1

Dear Dr. Oiwa,

We’re pleased to inform you that your manuscript has been judged scientifically suitable for publication and will be formally accepted for publication once it meets all outstanding technical requirements.

Kind regards,

Jennifer Annette Campbell, PHD, MPH

Academic Editor

PLOS ONE
---

## [Editor Report · Acceptance letter]

10 May 2023

PONE-D-22-23027R1 

Interval walking training in type 2 diabetes: A pilot study to evaluate the applicability as exercise therapy 

Dear Dr. Oiwa:

I'm pleased to inform you that your manuscript has been deemed suitable for publication in PLOS ONE. Congratulations! Your manuscript is now with our production department. 

Kind regards, 

on behalf of

Dr. Jennifer Annette Campbell 

Academic Editor

PLOS ONE